# Four Decades of *Bacillus* Biofertilizers: Advances and Future Prospects in Agriculture

**DOI:** 10.3390/microorganisms13010187

**Published:** 2025-01-17

**Authors:** Xinmai Wu, Yan Liu, Baolei Jia, Lili Tao, Han Li, Jingbang Wang, Ziqi Yuan, Xiaobao Sun, Yanlai Yao

**Affiliations:** 1Xianghu Laboratory, Hangzhou 311231, China; 17816767282@163.com (X.W.); jiabaolei@xhlab.ac.cn (B.J.); 15637470871@163.com (H.L.); wangjingbang2021@163.com (J.W.); yyuanzi77@163.com (Z.Y.); sunxiaobao@xhlab.ac.cn (X.S.); 2Jiangsu Provincial Key Lab of Solid Organic Waste Utilization, Jiangsu Collaborative Innovation Center of Solid Organic Wastes, Educational Ministry Engineering Center of Resource-Saving Fertilizers, Jiangsu Provincial Key Laboratory of Coastal Saline Soil Resources Utilization and Ecological Conservation, Nanjing Agricultural University, Nanjing 210095, China; 2020203065@stu.njau.edu.cn; 3Institute of Environment, Resource, Soil and Fertilizer, Zhejiang Academy of Agricultural Sciences, Hangzhou 310021, China

**Keywords:** *Bacillus*-based microbial fertilizers, sustainable agriculture, bibliometric, PGPR, synthetic microbial communities, multi-omics

## Abstract

Over the past four decades, *Bacillus* biofertilizers, which are microbial formulations based on *Bacillus* species, have significantly contributed to sustainable agriculture by enhancing crop growth, improving soil health, and reducing the dependency on chemical fertilizers. *Bacillus* species, particularly known for their ability to promote plant growth, fix nitrogen, solubilize phosphorus, and produce growth-promoting substances such as phytohormones and antibiotics, have emerged as key players in the development of eco-friendly agricultural solutions. This research utilizes bibliometric analysis based on 3,242 documents sourced from the Web of Science database to map the development, key contributions, and innovation within the field from 1985 to 2023. This study identifies exponential growth in research output, particularly from 2003 onwards, indicating a robust interest and expanding research base predominantly in China, India, and the United States. We segmented the research timeline into three distinct phases, each marked by varying growth rates and research foci. This paper presents novel insights into the geographical and institutional distributions of research, highlighting the predominant role of developing countries in advancing *Bacillus*-based technologies. Key research hotspots have evolved from basic applications to complex interactions involving synthetic microbial communities and advanced multi-omics techniques. Our findings demonstrate a trend towards more strategic and technologically integrated approaches to developing *Bacillus* biofertilizers, reflecting broader shifts towards more sustainable agricultural systems. This study not only charts historical progress, but also proposes future research trajectories aimed at enhancing the application and effectiveness of microbial fertilizers across diverse ecosystems.

## 1. Introduction

Agriculture is currently facing a series of severe challenges, including land resource scarcity, water shortages, worsening environmental pollution, drastic climate change, and food security issues [1,2,3]. The root causes of these problems lie in accelerated population growth, excessive resource exploitation, insufficient environmental protection, exacerbated global climate change, and deficiencies in management policies [4,5]. To address these challenges, agriculture urgently needs to transition toward sustainability and smart farming, improving resource utilization efficiency, reducing environmental pollution, and enhancing the resilience of agricultural systems to ensure food security and the long-term development of agriculture [6,7]. Microbial formulations have shown great potential in improving soil health, promoting crop growth, inhibiting pathogens, enhancing plant stress resistance, and reducing the dependence of agriculture on chemical inputs [8,9,10,11]. Their research and use are becoming increasingly widespread [12].

*Bacillus* preparations are a very important type of agricultural microbial formulation [13]. A variety of *Bacillus* preparations have been shown to positively regulate plant growth [14,15]. Strains of the genus *Bacillus* are bacteria capable of producing heat-resistant endospores, classified phylogenetically as bacteria, *Firmicutes*, *Bacilli*, *Bacillales*, and *Bacillaceae* [16]. Currently, this *genus* includes over 300 species [17]. The classification is quite complex, and, with ongoing research, many species in this genus have been reclassified into other genera, becoming closely related to the genera of *Bacillus*. Therefore, this article discusses all taxonomic groups under *Bacillaceae* that are functionally similar to *Bacillus* and can be used as agricultural microbial formulations. For convenience, they will all be collectively referred to as *Bacillus*. *Bacillus* is one of the common genera in the rhizosphere and is also a very important type of plant-growth-promoting rhizobacteria (PGPR), which has been extensively studied. They naturally exist in the rhizosphere, and their secreted metabolites can strongly influence the rhizosphere microenvironment [18]. Through various means, such as activating rhizosphere nutrients, producing multiple plant-growth-promoting substances, inhibiting and antagonizing plant pathogenic microorganisms, and inducing plant systemic resistance, they directly or indirectly promote plant growth [19,20].

*Bacillus* plays an important role in agricultural production, with wide applications and promising prospects, and has become an integral part of sustainable agricultural system management [21,22]. In the United States, 60% to 75% of crops such as cotton, peanuts, soybeans, corn, vegetables, and grains are treated with commercial *Bacillus* products, effectively combating soil-borne pathogens like *Fusarium* spp. and *Rhizoctonia* spp. [23]. In China, 80% of registered biofertilizer strains belong to *Bacillus* spp. (http://www.biofertilizer95.cn/wsw_djcp/index.htm, accessed on 15 July 2024). The *Bacillus* industry is developing rapidly and thriving. *Bacillus* products for agricultural use are diverse, and this article focuses on their research as biofertilizers. This article relies on the Web of Science database and employs bibliometric analysis to statistically analyze academic research papers related to *Bacillus*-based agricultural formulations, aiming to reveal development trends, current status, research hotspots, and other aspects in this field. It provides strong guidance and support for future research directions, academic collaborations, resource allocation, and strategic decision making.

## 2. Materials and Methods

### 2.1. Data Collection

The data were collected from the Web of Science core collection database. The specific retrieval formula was Topic Search (TS) = (“*Bacillaceae*” OR “**bacillus**” NOT “*lactobacillus**”) AND TS = (“inoculant” OR “manure” OR “fertiliz*” OR “fertilis*” OR “biofertiliz*” OR “biofertilis*” OR “bio-fertiliz*” OR “bio-fertilis*”) AND TS = (“plant*” OR “soil*” OR “grow*” OR “agric*” OR “agrono*” OR “rhizosphere” OR “PGPR” OR “PGPB” OR “PGPM” OR “plant* growth* promot*” OR “plant-growth* promot*”) NOT TS = (“ensilage” OR “ensiling” OR “silage”).

### 2.2. Data Analysis

Excel (v.2016) was used to organize the data, Origin (v.2024) to draw the bar plot and perform the fitting (the fitting model is ExpGrow1), VOSviewer (v.1.6.20) to perform a visual collinear analysis of countries and keywords [24], and Bibliometrix (v.4.3.0) to analyze the top 10 articles [25].

## 3. Results

### 3.1. Number of Articles Related to Bacillus Biofertilizers

The relevant papers were searched from 1985 to 2024, but because the paper data collection for 2024 was incomplete, the papers from 1985 to 2023 were selected as the analysis object. As of December 2023, a total of 3242 papers were retrieved, including 2975 articles (accounting for about 91.76%), 175 reviews (about 5.40%), 115 proceedings papers (about 3.55%), and 17 other types of papers (about 0.52%) (Appendix A). The most commonly used language is English, accounting for about 98.19%, followed by Portuguese, but only accounting for about 0.55% (Appendix A). This shows that research work related to *Bacillus* biofertilizers is mainly published in English journals, and most of them are research papers, but there are also some critical opinion papers. Next, we mainly analyzed 2975 article-type papers. During these 38 years, the number of articles published showed an exponential growth trend (Figure 1). According to the growth trend, the research is divided into the following three stages: from 1985 to 2002, the growth was slow, from 5 articles to a peak of 14 articles (2000); from 2003 to 2014, the growth accelerated, from 19 articles to 89 articles; and from 2015 to 2023, the growth was rapid, from 124 articles to 443 articles (2022), an increase of nearly 2.6 times. This shows that *Bacillus* biofertilizers are increasingly valued by researchers around the world.

### 3.2. Countries, Institutions, and Senior Scholars

We counted the number of articles from different countries (Figure 2A) and showed the top 10 countries (Appendix A), with a total of 2494 articles, accounting for 83.83%. China is the country with the most articles (896), followed by India (417), the United States (326), Pakistan (193), Brazil (183), South Korea (104), Egypt (97), Saudi Arabia (93), Spain (93), and Germany (92). The top 7 countries (Figure 2B) have a total of 2216 articles, accounting for 74.49%. The number of articles in all countries showed a gradual increase, with China and India being the most obvious. Among these 7 countries, only the USA and South Korea are developed countries, and the remaining 5 countries are developing countries. This shows that developing countries have more demand and research for *Bacillus* biofertilizers. There is friendly cooperative research between different countries. Statistics show that China is the country with the most co-authored articles (336 articles), followed by the USA (209), Pakistan (189), Saudi Arabia (186), India (145), Egypt (119), Germany (108), Australia (78), Spain (70), Italy (62), etc. (Figure 2C). The cooperation between China and the USA, China and Pakistan, Pakistan and Saudi Arabia, and Saudi Arabia and Egypt is the closest. Collaborative publishing not only enhances the quality and depth of research, but also aids in the personal career development of researchers and advances scientific knowledge. International collaboration, in particular, enables researchers to expand their academic horizons, play a more active role in the global scientific community, and collectively push scientific progress forward.

We studied the institutions ranked in the top 20 for publishing articles (Appendix A). Seven of them are in China, three in the USA, three in Brazil, two in India, one in Egypt, one in Saudi Arabia, one in Turkey, one in Pakistan, and one in Canada. The top five institutions are Nanjing Agricultural University in China (157 articles), Chinese Academy of Sciences (128), Ministry of Agriculture Rural Affairs (128), Indian Council of Agricultural Research Icar in India (111), and Egyptian Knowledge Bank Ekb in Egypt (108).

Understanding the senior scholars in a research field not only guides research directions and methods, but also promotes academic collaboration and enhances research impact. We summarized the top 20 scholars in the field of *Bacillus* biofertilizers based on publication rankings (Appendix A). Among them, ten are Chinese scientists, all affiliated with Nanjing Agricultural University; two are Brazilian scientists, both affiliated with the Federal University of Pernambuco; three are Pakistani scientists, affiliated with the University of Haripur, the University of Agriculture Faisalabad, and Quaid-i-Azam University; three are Polish scientists, affiliated with Wrocław University of Environmental and Life Sciences and Warsaw University of Life Sciences; two are Turkish scientists, affiliated with Ondokuz Mayis University and Yeditepe University; and one is an American scientist, affiliated with Auburn University. These 20 scientists’ research fields cover soil science, microbial ecology, plant pathology, environmental biotechnology, plant physiology, microbial fertilizers, and more, reflecting a broad focus on the interactions between *Bacillus* and plants in the fields of agriculture and environmental science.

### 3.3. Journal, Research Areas, and Most Highly Cited Articles in the Field

We were also interested in the journals that published on *Bacillus* biofertilizers. Therefore, we performed a statistical analysis and showed the top 20 journals in terms of the number of articles (Appendix A). Among these journals, five journals were founded in Switzerland, four journals in the Netherlands, four journals in the UK, three journals in Germany, three journals in the USA, and one journal in Chile. In 2023, their impact factors ranged from 1.300 to 10.753, and their CiteScores ranged from 2.7 to 20.8. Frontiers in Microbiology is the journal with the highest number of publications on *Bacillus* biofertilizers (99 articles), followed by Agronomy-Basel (86) and Microorganisms (64), all of which are based in Switzerland. Next are Applied Soil Ecology (51) and Scientia Horticulturae (46), both of which are based in the Netherlands. Frontiers in Microbiology, founded in 2010, covers all fields of microbiology, emphasizing innovative and high-quality research, and offers free open access. Agronomy-Basel, founded in 2011, focuses on the science and technology of crop production and utilization, with a commitment to sustainable agriculture research, and also provides free open access. It covers areas such as crop physiology, soil fertility, and crop production. Microorganisms, founded in 2013, covers a wide range of microbiology-related topics, from basic research to applications, and offers free open access. Its content includes microbial genetics, microbial interactions, and microbial biotechnology, among other fields. Applied Soil Ecology, founded in 1994, is an academic journal focused on soil ecology. The journal is dedicated to studying the complex interactions and functions of soil ecosystems, promoting sustainable land management and environmental protection, and provides a free open-access platform. Scientia Horticulturae, founded in 1973, is an academic journal focused on horticultural science. The journal aims to advance research on the production, management, and utilization of horticultural crops, and provides a free open-access platform, supporting the development of horticultural science worldwide.

Understanding the research fields of journals where articles are published not only helps in evaluating the practical value and impact of research on *Bacillus* biofertilizers, but also guides future research directions, supports policy and decision making, promotes commercialization and technology transfer, and enhances social and economic benefits. Consequently, we summarized the research fields of these articles and ranked them based on the number of articles (Appendix A). The most represented field is Agriculture (999 articles), followed by Environmental Sciences Ecology (593), Microbiology (564), and Plant Sciences (547). Other significant fields include Biotechnology Applied Microbiology (412), Science Technology Other Topics (209), Engineering (412), and Chemistry (137). This demonstrates the intersection and integration between various fields.

The citation count of an article reflects its importance, impact, and broad applicability in the academic community. Among the top 10 most-cited articles, four are from the USA, three from China, and the remaining three from the UK, Italy, and Canada, with publication years ranging from 1997 to 2015 (Table 1). The top-ranked article, cited 492 times, studied the decisive impact of soil pH on microbial diversity and composition [26]. The second-ranked article, cited 469 times, discussed how plant-growth-promoting rhizobacteria can reduce plants’ need for chemical fertilizers [27]. The third-ranked article, cited 435 times, evaluated the effects of biofertilizers, including nitrogen-fixing bacteria, phosphate- and potassium-solubilizing bacteria, and arbuscular mycorrhizal fungi, on maize growth [28]. The fourth-ranked article, a review cited 432 times, introduced the issue of pathogen contamination in fresh produce due to improperly treated compost [29]. The fifth-ranked article, cited 407 times, studied and revealed the structure of urease from *Bacillus pasteurii* in its native and inhibited states, providing a new mechanism for enzymatic urea hydrolysis [30]. These 10 articles cover various fields, including soil microbial diversity, plant-growth-promoting rhizobacteria, biofertilizers, enzyme mechanisms, soil microbial biomass and activity, biofilms, phosphate-solubilizing bacteria in the rhizosphere, the role of rhizosphere microorganisms in drought resistance, and the effects of continuous cropping on microbial communities. These articles showcase significant research achievements in microbial ecology, soil biochemistry, and environmental microbiology, with profound implications for the development of agricultural and environmental sciences.

### 3.4. Keywords of Articles

To better understand the research hotspots of *Bacillus* biofertilizers over the past 38 years, we conducted a keyword analysis of the downloaded articles. Synonymous keywords were consolidated based on research experience. There are a total of 6453 keywords in these articles, with 42 keywords appearing more than 30 times (Figure 3A). These 42 keywords account for only 58.71% (Appendix A), indicating a high proportion of low-frequency keywords and suggesting that research directions on *Bacillus* biofertilizers are complex and diverse. The top 15 keywords by frequency are PGPR (393 times), biofertilizer (361), plant growth (286), *Bacillus* (237), phosphorus (281), microbial community (200), biological control (122), *Bacillus subtilis* (108), rhizosphere (106), endophyte (98), compost (97), fertilizer (95), maize (73), *Pseudomonas* (71), and yield (69). The combined proportion of the top 15 keywords is only 39.27%, indicating that the frequency of the top keywords is relatively even, implying a high correlation between various research directions of microbial fertilizers. Additionally, the higher the frequency of a keyword, the higher its total link strength, indicating that the keyword is a focus, or even a central point, in various research directions.

To clarify the changes in research hotspots of *Bacillus* biofertilizers, we analyzed the keywords for different periods. From 1985 to 2002, there were 292 keywords, with 40 keywords appearing more than twice (Figure 3B). From 2003 to 2014, there were 1523 keywords, with 37 keywords appearing more than eight times (Figure 3C). From 2015 to 2023, there were 5343 keywords, with 36 keywords appearing more than 27 times (Figure 3D). It can be seen that the number of keywords and research frequency have greatly increased. Based on the content of the keywords, they can be roughly divided into seven categories: “Microorganism”, “Application”, “Plant”, “Nutrition”, “Location”, “Mechanisms”, and “Methods”. In different periods, there were also other categories of keywords such as “Animals”, “Environment”, and “Other”. Combining these categories, we can clearly see that the past research hotspots of *Bacillus* biofertilizers generally focus on using beneficial microorganisms (“Microorganism”) to develop agricultural inputs (“Application”), and promoting plant (“Plant”) nutrient absorption (“Nutrition”) and healthy growth. Different research methods (“Methods”) are used to explore the mechanisms (“Mechanisms”) by which beneficial microorganisms promote plant growth and disease resistance in different habitats (“Location”). This also includes the impact on soil animals (“Animals”), responses and improvements to adverse environments (“Environment”), and the significance for sustainable agricultural development (“Other”).

The main categories of research hotspots have not changed significantly over the 38 years, but the specific research subjects within these categories have changed. The number of beneficial microbial species has increased, and research has gradually shifted from single strains and single species to microbial community research. Correspondingly, research methods for microbial communities have also been innovated and changed, transitioning from low-throughput time-consuming methods (DGGE, Denatured Gradient Gel Electrophoresis) [36] to the precise and efficient high-throughput sequencing era [37]. Research on the “Application” of microorganisms has included biostimulant, products centered on microbial metabolites, on top of traditional biological control, biofertilizers, and compost products. Research on “Plant” remains focused on promoting the growth, disease control, and yield assurance of crops, but research on economic crops (tomato) has gradually increased beside staple crops (wheat, maize, rice). In the field of “Nutrition”, the availability of soil nutrient elements and plant nutrient absorption remain evergreen topics, with phosphorus and nitrogen receiving widespread attention from soil scientists and plant nutritionists. The research on the “Location” of beneficial microorganism and plant interactions has gradually shifted from soil to rhizosphere and then to endophyte, made possible by advancements in natural science research methods, allowing for macro to micro studies. Research on the “Mechanisms” of beneficial microorganism and plant interactions mainly focuses on soil enzymes and plant hormones, especially the plant growth hormone IAA, which is a major research focus. Additionally, with the aging of the earth and increased human activities, various environmental issues have emerged, such as salinity stress, making sustainable agricultural development a hot topic and key research subject.

### 3.5. Research Content on PGPR Bacillus

The study of *Bacillus* species as PGPR has become a major research focus in the current agricultural field. However, analyzing only the top 36 keywords from 2015 to 2023 cannot clearly present the main research themes and development trends regarding *Bacillus* as PGPR. Therefore, we analyzed all keywords from these nine years and categorized and summarized them to clearly reveal the current research content and trends of *Bacillus* as PGPR (Figure 4).

The research hotspots derived from keyword analysis in studies on *Bacillus* biofertilizers published during 2015–2023 underscore key focus areas such as plant growth enhancement, soil quality improvement, and the mitigation of environmental stresses like salinity and drought. Advanced methodologies such as genomics, transcriptomics, and metabolomics are highlighted for their role in deepening our understanding of *Bacillus* functions within agricultural settings. This encapsulates the extensive scope of research themes, showcasing the essential role of *Bacillus* in promoting sustainable agricultural practices and addressing the challenges of modern farming.

Due to various factors such as natural climate, global change, and human activities, modern agricultural production faces numerous challenges, including antibiotic contamination [38], salinity [39], heavy metal [40], drought [41], continuous cropping obstacles [42], chemical fertilizer abuse [43], plant diseases [44], and soil quality [45]. Microbial formulations, primarily microbial fertilizers, developed with *Bacillus* species as the core functional strains of PGPR, are focused on addressing these challenges while promoting plant growth, ensuring and enhancing crop yields, improving fruit quality, and maintaining agricultural sustainability [46,47].

The hot research technologies for PGPR *Bacillus* as agricultural inputs can be broadly summarized into five points: SynComs, genome, microbiome, transcriptome, and metabolome. Given the functional limitations of single strains, synthetic microbial communities (SynComs) has been developed and applied in agricultural production. SynComs can simulate the complexity of natural microbial communities, provide diverse biological functions, and serve as model systems to study the assembly principles and dynamic changes in microbial communities. This approach helps to elucidate the complex interactions between PGPR and host plants [48]. In recent years, with the rise in multi-omics technologies such as genomics, microbiomics, and transcriptomics, the “black box” of the plant microbiome is gradually being unveiled [49,50]. SynComs (Synthetic Microbial Community), combined with multi-omics technologies, allows for the in-depth analysis of SynCom functions, PGPR–plant interactions, environmental adaptability, and the optimization of SynCom design [39,51,52].

In addition to the aforementioned hot research techniques, we summarized 14 other research focal points for PGP *Bacillus* which include root exudates, colonization, biofilm, biological N fixation, phosphorus solubilization, phytohormones, amino acids, ACC deaminase (1-aminocyclopropane-1-carboxylate deaminase), antibiotics, VOCs (Volatile Organic Compounds), antioxidants, ISR (Induced Systemic Resistance), ARGs (Antibiotic Resistance Genes), and cellulose degradation. These topics cover the interactions between *Bacillus* and plants, the mechanisms through which *Bacillus* promotes plant growth and stress resistance, the relationship between *Bacillus* and the spread of antibiotic resistance genes, and the application of *Bacillus* in the resource recovery of straw and other solid wastes. These research focal points demonstrate the multifunctionality, diversity, and depth of research on *Bacillus* in the agricultural sector, highlighting its potential in the development of biofertilizers, economic growth in agriculture, and environmental protection. This breadth of research offers innovative perspectives and methods for developing new agricultural technologies, thus supporting the sustainability of agricultural production and long-term environmental stability.

## 4. Discussion

Based on the results of our analysis, we will engage in a thorough discussion on the mechanisms of plant–*Bacillus* interactions and explore the future role of *Bacillus* biofertilizers in agriculture.

### 4.1. Plants Recruiting PGPR Bacillus

Root exudates have been identified as a key regulatory mechanism in plant–PGPR interactions [53,54]. Plants secrete root exudates into the soil to recruit PGPR *Bacillus* to aid in their growth and stress resistance [55,56]. The types and abundance of root exudates vary among different plant species, different growth stages, and conditions of the same plant [57]. This variation determines the plant’s ability to selectively recruit microbial communities, thereby optimizing its growth and stress resistance under different environmental conditions. This is also a key driving factor for the diversity and function of the plant microbiome. PGPR *Bacillus* senses the plant’s root exudates through its receptors, moves to the plant rhizosphere by flagellar rotation, and eventually colonizes the root surface or rhizosphere region to exert its plant-growth-promoting functions [58,59]. The ability of PGPR *Bacillus* to form biofilms is crucial for its colonization at the plant roots [60]. Biofilms are complex structures composed of polysaccharides, proteins, and other biological macromolecules secreted by bacteria, providing a protective environment that enables bacteria to stably colonize plant roots [61]. Previously, the “rhizosphere microbiome” was referred to as the plant’s second genome [62]. However, with increasing research on microbiomes in other plant parts such as endophytic and phyllospheric communities [63,64], it appears more appropriate to use the term “plant microbiome”. *Bacillus*, although not a high-abundance dominant species within the plant microbiome, occupies a critically important position due to its multiple beneficial traits [65].

### 4.2. Plant-Growth-Promoting Mechanisms of Bacillus

As a functional strain in microbial fertilizers, *Bacillus* can directly or indirectly promote plant growth through its diverse mechanisms, including enhancing nutrient absorption, improving plant stress resistance, inhibiting pathogenic microorganisms, and regulating plant hormone levels, thereby significantly improving plant growth and development, disease resistance, and environmental adaptability.

(1) Biological nitrogen fixation. The genomes of many *Bacillus* strains contain nitrogen fixation genes, such as *nifB*, *nifH*, *nifD*, *nifK*, *nifE*, and *nifN* [66], which confer non-symbiotic nitrogen-fixing abilities to *Bacillus*. Rhizosphere probiotics such as *Bacillus subtilis*, *Bacillus pumilus*, *Bacillus megaterium*, *Bacillus circulans*, *Bacillus firmus*, *Bacillus cereus*, *Bacillus licheniformis*, *Bacillus mycoides*, and *Paenibacillus polymyxa* can assist in nitrogen fixation for plants, alleviating nitrogen starvation [67,68,69,70]. Biological N fixation is considered one of the most promising technologies to replace chemical nitrogen fertilizers as the primary nitrogen source for food crops [71]. Enhancing the efficiency of nitrogen-fixing microorganisms and expanding their application, particularly in non-leguminous crops such as rice, wheat, and maize, will contribute to reducing chemical nitrogen fertilizer usage and achieving the goal of sustainable food production.

(2) Phosphorus solubilization. *Bacillus* strains secrete organic acids (such as citric acid, acetic acid, oxalic acid, etc.) and enzymes like phosphatases to convert insoluble phosphates in the soil (such as tricalcium phosphate, aluminum phosphate, and iron phosphate) into soluble phosphates [72,73]. Its phosphorus-dissolving effect is affected by factors such as phosphate source, temperature, and organic carrier [74,75,76]. The application of microbial fertilizers containing phosphorus-solubilizing *Bacillus* can significantly increase crop yields across various crops [77,78]. This not only improves the availability of phosphorus nutrients in the soil, but also reduces dependence on chemical phosphorus fertilizers, helping to lower agricultural production costs and reduce environmental pollution [79]. The internal molecular mechanisms of phosphate-solubilizing *Bacillus*, the screening and genetic modification of high-efficiency phosphate-solubilizing *Bacillus* strains, and the interaction mechanisms between phosphate-solubilizing *Bacillus* and plants are current research hotspots [80,81]. Future studies should focus more on multi-omics integration, environmental adaptability, and the synergistic effects within microbial communities to promote the widespread application of phosphate-solubilizing *Bacillus* in agriculture.

(3) Phytohormones. PGPR *Bacillus* produce various plant hormones, such as auxins, cytokinins, and gibberellins, to regulate plant growth and development [82,83,84]. By secreting these plant hormones, PGPR *Bacillus* can significantly influence the growth and development of plants, enhance their adaptability to adverse environmental conditions, promote healthy growth, and ultimately improve crop yield and quality [85]. *Bacillus subtilis* promotes soybean growth through the production of auxin [86]. *Bacillus cereus*, *Bacillus macrolides*, and *Bacillus pumilus* enhance red pepper growth by producing gibberellin [87]. Additionally, *Bacillus* species promote plant growth and enhance stress tolerance by inducing and regulating the production and transport of endogenous hormones in plants. *Bacillus endophyticus* induces the increase in auxin, cytokinin, and gibberellin levels in plants through stress response, thereby effectively alleviating osmotic stress [88]. Psychrophilic *Bacillus* spp. can positively regulate the expression of phytohormones and significantly improve the growth of wheat under cold stress [89]. *Bacillus amyloliquefaciens* positively modulated the expression of stress-responsive genes under salinity and alter phytohormone levels in *OsNAM*-overexpressed plants [90]. Currently, research on *Bacillus*-producing hormones and promoting plant growth is extensive. However, due to limitations in technical methods, such as inaccurate mass spectrometry predictions and incomplete metabolite databases, a significant gap remains in the study of plant hormone analogs from other microbial sources.

(4) ACC deaminase. PGPR *Bacillus* secretes ACC deaminase to lower ethylene levels in plants, helping them maintain normal growth and development under stress conditions, thereby improving the stability and sustainability of agricultural production [91,92,93]. *Bacillus licheniformis* HSW-16, which produces ACC deaminase, can improve salt stress in wheat plants [94]. *Bacillus amyloliquefaciens* MMR04 and *Bacillus subtilis* MMR18 produce ACC deaminase, which degrades the ethylene produced by *Pennisetum glaucum* under drought stress, thereby protecting its healthy growth [95]. In addition, four strains capable of producing ACC deaminase (*Bacillus subtilis* PM32Y, *Bacillus cereus* WZ3S1, *Bacillus* SM73, and *Bacillus* WZ3S3) were used in association with alfalfa to perform phytoremediation on petroleum-hydrocarbon-contaminated soil, achieving significant remediation effects [96]. Current research hotspots primarily focus on elucidating the regulatory mechanisms of ACC deaminase, as well as screening and optimizing high-efficiency strains for enhanced performance [97,98]. In the future, research is likely to focus on the development of more efficient and stable *Bacillus* strains with enhanced ACC deaminase production, as well as the exploration of their synergistic effects with other beneficial microorganisms.

(5) Antibiotics. Several species of the genus *Bacillus* produce antibiotics, which are synthesized either through a ribosomal or non-ribosomal mechanism [99]. Antibiotics synthesized via non-ribosomal pathways are one of the key features of *Bacillus* in antagonizing pathogenic microorganisms. These include cyclic lipopeptide compounds such as surfactin, iturins, fengycin, and bacillibactin, as well as polyketide compounds like difficidin, macrolactin, and bacillaene [61,100,101,102]. *Bacillus amyloliquefaciens* S13-3 produces lipopeptide antibiotics that reduce the severity of strawberry anthracnose caused by *Colletotrichum gloeosporioides* [103]. *Bacillus subtilis* CAS15 has also shown strong inhibitory effects against 15 different pathogens, including *Fusarium* sp., *Bacillus anthracis*, *Pythium* sp., *Trichoderma* sp., and *Phytophthora* sp., due to the production of catechol-type siderophores [104]. Research has reported that 10% of the *Bacillus velezensis* FZB42 genome (340 kb) is dedicated to the synthesis of lipopeptides, polyketides, siderophores, and bacteriocins [105]. The secretion of antibiotics by *Bacillus* makes it a powerful biocontrol agent, effectively inhibiting or killing pathogens, reducing the occurrence of plant diseases, and decreasing the use of chemical pesticides, thereby contributing to more sustainable and environmentally friendly agricultural production. As a result, many *Bacillus*-based biocontrol agents for plant disease resistance have already been developed and applied in the market, demonstrating good effectiveness and promising prospects.

(6) VOCs. PGPR *Bacillus* produces VOCs such as 2,3-butanediol, acetic acid, acetoin, dimethyl disulfide, and phenylacetic acid, which promote plant growth and enhance plant resistance [106,107,108]. *Bacillus subtilis* GB03, *Bacillus amyloliquefaciens* IN937a, and *Bacillus velezensis* SQR9 release a mixture of volatile compounds that promote the growth of *Arabidopsis*. The identified components include acetoin and 2,3-butanediol, among others [109,110]. A mixture of volatile compounds containing acetoin produced by *Bacillus amyloliquefaciens* UCMB5033 exhibited strong and significant growth inhibition against fungal pathogens (*Botrytis cinerea*, *Alternaria brassicicola*, *Alternaria brassicae*, *Sclerotinia sclerotiorum*) [111]. Currently, there is considerable research on VOC mixtures with plant probiotic functions. However, progress in studying individual compounds has been slow due to the cumbersome screening processes, incomplete metabolite databases, the instability of substances, and the difficulty in obtaining pure compounds. There is an urgent need for new technologies to overcome these bottlenecks [112].

(7) Antioxidant. PGPR *Bacillus* stimulates plants under stress conditions to produce antioxidants, including antioxidant enzymes like superoxide dismutase (SOD), catalase (CAT), peroxidase (POD), and ascorbate peroxidase (APX) [113,114], as well as reducing agents such as glutathione (GSH) [115], ascorbic acid (vitamin C) [116], and phenolic compounds [117]. These substances help plants regulate the antioxidant system to reduce oxidative damage caused by stress, improving their resistance and survival ability. For example, *Bacillus anthracis* PM21 promotes the growth of corn plants under salt stress by increasing the activities of APX, SOD, POD, and CAT [118]. Similar mechanisms have been observed in other crops such as cucumber, soybean, and wheat [119,120,121]. In the context of increasing environmental pressures, *Bacillus* biofertilizers provide practical solutions for the ecological restoration of harsh environments and sustainable agricultural development, and show broad application prospects.

(8) ISR. ISR-inducing *Bacillus* strains can trigger the plant’s internal defense signaling network, activating broad-spectrum disease and stress resistance responses, thereby protecting plants from pathogenic microbial infections [20,122,123]. *Bacillus cereus* NJ01 can induce immune responses in rice, protecting it against infection by the pathogen *Xanthomonas oryzae* [124]. *Bacillus amyloliquefaciens* induces the systemic resistance of perennial ryegrass to *Magnaporthe oryzae* through surfactin [125]. Immune-inducing *Bacillus* strains are environmentally friendly and have a low risk of developing resistance, making them a promising green biopesticide for plant disease control and crop yield enhancement [126]. However, only a few *Bacillus* strains have been extensively studied and applied in agriculture so far. There is a lack of efficient strains that can adapt to different environmental and crop conditions, and the interaction mechanisms between *Bacillus* and various plants are not yet fully understood. Additionally, there is a shortage of long-term field trials to assess the risks, and the development of *Bacillus* as commercial products also faces technical and market promotion challenges.

(9) Others. In addition to the growth-promoting mechanisms mentioned in the above keywords, *Bacillus* can also exert its growth-promoting effects through a variety of other pathways. *Bacillus* can secrete metabolites such as organic acids, ammonia, and surfactants to regulate soil pH, thereby improving the soil environment and promoting plant growth [127,128,129]. *Bacillus* can also secrete a variety of enzymes (invertase, cellulase, and carbonic anhydrase) to accelerate the decomposition of organic carbon, promote nutrient cycling, and improve soil fertility [65,127]. By secreting substances such as organic acids, enzymes, and extracellular polysaccharides, *Bacillus* can dissolve insoluble potassium in the soil and improve the efficiency of potassium utilization by plants [130,131,132]. In addition, *Bacillus* also promotes plant growth by synthesizing siderophores, and the iron competition effect produced by siderophores can effectively inhibit pathogens [133,134]. After entering the soil, *Bacillus* can reshape the structure of soil microbial communities by enriching beneficial microorganisms and reducing the abundance of pathogens, thereby improving the disease resistance and overall growth of plants [135,136,137]. These multifaceted functions highlight the potential of Bacillus as a promising biofertilizer and biocontrol agent.

### 4.3. Future Research Directions

There are many areas where future research on the agricultural applications of PGPR *Bacillus* needs to be intensified, including the following:

(1) Strain selection and improvement: Utilizing genomics and metabolomics approaches to select more efficient strains of *Bacillus* that could potentially be passed along with plant seeds, or through genetic engineering to enhance existing strains for better performance and stability under various soil and environmental conditions.

(2) Development of application technologies: Developing new application techniques and formulations, such as microencapsulation or combination formulations with other beneficial microbes, to improve the survival and effectiveness of *Bacillus*.

(3) Effectiveness under multiple stress conditions: Investigating the survival and growth-promoting effects of *Bacillus* under various stress conditions such as saline–alkaline soils, drought, or excessively wet soils, and how it modulates plant physiological responses to adapt to these stresses.

(4) Analysis of molecular mechanisms: Conducting in-depth studies on the molecular signaling recognition and transmission mechanisms involved in the interaction between *Bacillus* and plants. Particularly, how it affects the plant’s immune system, balance of growth hormones, and capability to handle stress.

(5) Long-term application and ecological impact: Researching the impact of the long-term application of *Bacillus* formulations on soil microecology and its long-term effects on biodiversity, ensuring its environmental friendliness and sustainable use.

(6) Quantitative assessment and model development: Establishing mathematical models and simulation systems to quantitatively assess the growth promotion and disease resistance effects of *Bacillus* to optimize application strategies.

### 4.4. Limitations of This Study

Although we employed bibliometric methods to analyze research on *Bacillus* as a microbial fertilizer in agricultural applications, there are still some unavoidable limitations. First, our literature sources were limited to the Web of Science core collection database, which, while representative, is not comprehensive. Second, our analysis was restricted to articles published in English, thus presenting certain linguistic limitations. Additionally, due to variations in keyword expressions across different articles, we relied on author experience for identification and unification, which inevitably led to some omissions. Moreover, due to space limitations, many significant research findings could not be discussed in detail. Despite these limitations, we believe that this study still provides valuable periodic summaries, forward-looking perspectives, and scientific guidance for the field.

## 5. Conclusions

This study utilized bibliometric methods to comprehensively analyze the literature related to *Bacillus* biofertilizers in the agricultural sector. It presented trends in publication volume, key countries and institutions, researchers, journals, highly cited articles, and the distribution of keywords. By interpreting and discussing these findings, this study extracted the current core research hotspots in the field of *Bacillus*-based microbial fertilizers, providing valuable references and guidance for future research directions in this area.

## Figures and Tables

**Figure 1 microorganisms-13-00187-f001:**
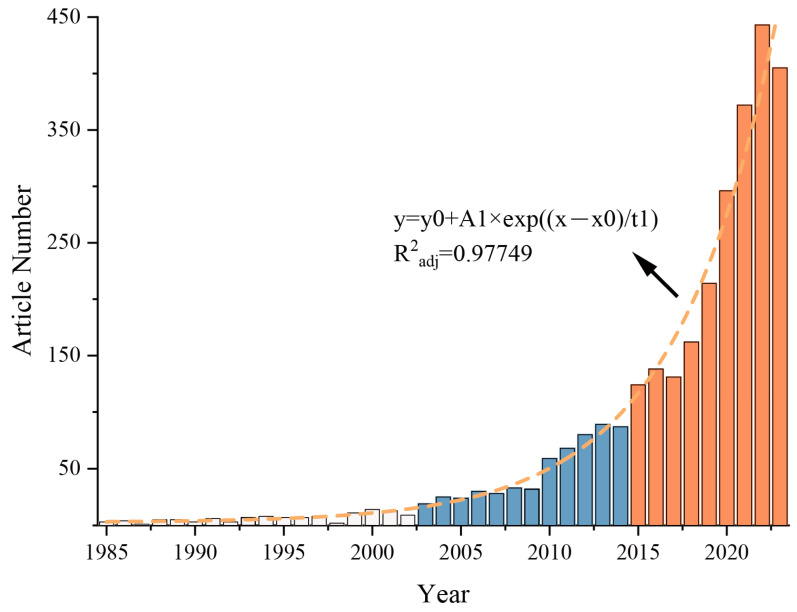
Annual number of articles in the field of *Bacillus* biofertilizers from 1985 to 2023.

**Figure 2 microorganisms-13-00187-f002:**
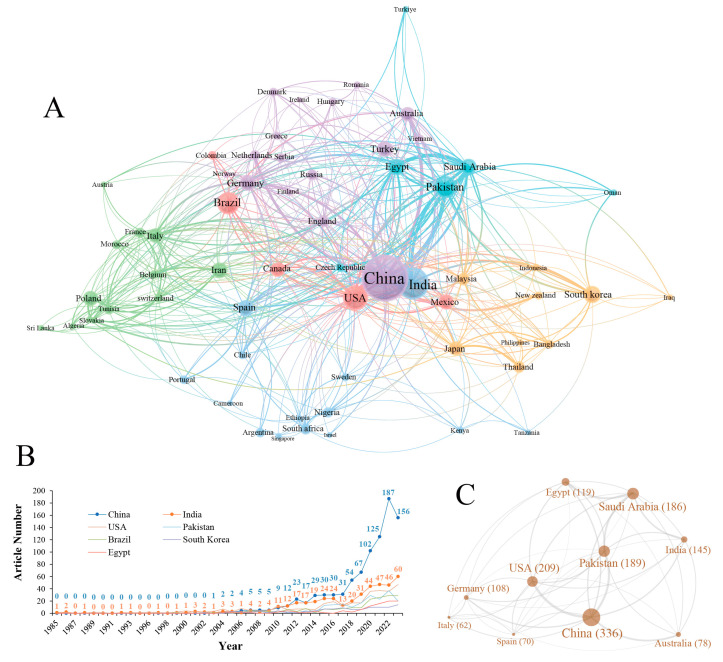
Publication trends of *Bacillus* biofertilizer articles from 1985 to 2023 across different countries. (**A**) Publication data for the top 65 countries ranked by volume, including their collaboration networks. Larger nodes indicate a higher number of publications from that country. Wider edges represent more frequent collaborations between countries. (**B**) Yearly publication trends for the top 7 countries. Numbers above each line indicate the number of publications for that year. (**C**) Top 10 countries ranked by the number of collaborative publications. Larger nodes indicate more collaborative publications, while wider edges represent more extensive collaborations.

**Figure 3 microorganisms-13-00187-f003:**
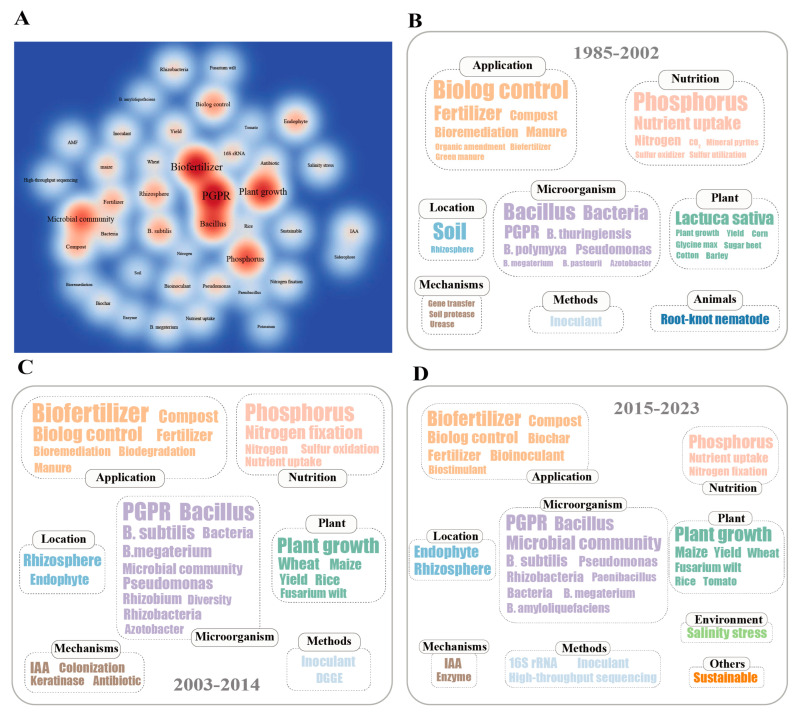
Distribution and trends of keywords in *Bacillus* biofertilizer research from 1985 to 2023. (**A**) A density map of 42 keywords that appeared more than 30 times during 1985–2023. Darker red indicates a higher frequency of occurrence. (**B**) A word cloud of 40 keywords that appeared more than twice from 1985 to 2002. (**C**) A word cloud of 37 keywords that appeared more than eight times from 2003 to 2014. (**D**) A word cloud of 36 keywords that appeared more than 27 times from 2015 to 2023. Keywords of the same color are categorized into the same group in the word cloud.

**Figure 4 microorganisms-13-00187-f004:**
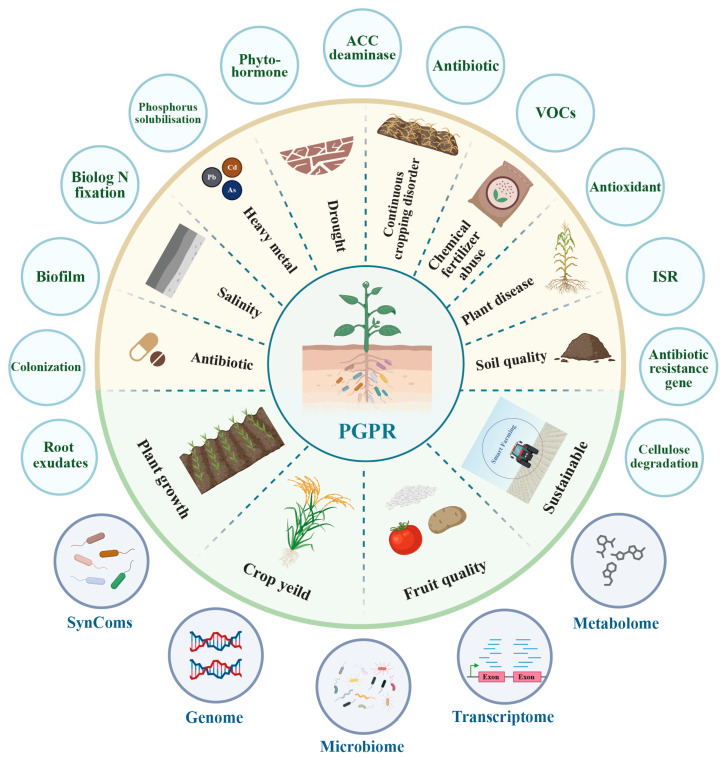
Research hotspots in *Bacillus* biofertilizers from 2015 to 2023.

**Table 1 microorganisms-13-00187-t001:** Top 10 most-cited articles in the field between 1985 and 2023.

No.	Article Title	Total Citations	TC per Year	Normalized TC	Country	Journal	Reference
1	Soil pH determines microbial diversity and composition in the park grass experiment	492	49.2	11.40	UK	MICROB ECOL	[26]
2	Plant-growth-promoting rhizobacteria allow for reduced application rates of chemical fertilizers	469	29.31	9.52	USA	MICROB ECOL	[27]
3	Effects of biofertilizer containing N-fixer, P and K solubilizers, and AM fungi on maize growth: a greenhouse trial	435	21.75	8.47	China	GEODERMA	[28]
4	Produce handling and processing practices	432	15.43	3.99	USA	EMERG INFECT DIS	[29]
5	A new proposal for urease mechanism based on the crystal structures of the native and inhibited enzyme from *Bacillus pasteurii*: why urea hydrolysis costs two nickels	407	15.65	7.69	Italy	STRUCTURE	[30]
6	Soil microbial biomass, dehydrogenase activity, and bacterial community structure in response to long-term fertilizer management	378	21.00	6.48	China	SOIL BIOL BIOCHEM	[31]
7	*Bacillus subtilis* biofilm induction by plant polysaccharides	366	30.50	9.36	USA	P NATL ACAD SCI USA	[32]
8	Phosphate-solubilizing rhizobacteria enhance the growth and yield but not phosphorus uptake of canola (*Brassica napus* L.)	360	12.86	3.33	Canada	BIOL FERT SOILS	[33]
9	Improved plant resistance to drought is promoted by the root-associated microbiome as a water-stress-dependent trait	355	35.50	8.22	USA	ENVIRON MICROBIOL	[34]
10	Different continuous cropping spans significantly affect microbial community membership and structure in a vanilla-grown soil as revealed by deep pyrosequencing	340	34.00	7.88	China	MICROB ECOL	[35]

## Data Availability

The original contributions presented in this study are included in the article/Appendix A. Further inquiries can be directed to the corresponding authors.

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
