# Peer review of "Four Decades of Bacillus Biofertilizers: Advances and Future Prospects in Agriculture"

_microorganisms, 2025, doi:10.3390/microorganisms13010187_

Round 1

Reviewer 1 Report

Comments and Suggestions for Authors

Four decades of Bacillus biofertilizers: advances and future prospects in agriculture

Abstract

Authors need to deep the concept of “Bacillus biofertilizers” or/and “Bacillus”

Keywords:

Use significant words but different from those in title.

Methodology

Ok

Results

See comments in MS.

Figure 3 It is difficult to see and, therefore, to read. Font size, in the density map, does not allow reading the smallest names. The same occurs with the colors. See comment in MS.

Discussion

Ok

Conclusion

Ok

Reviewer 2 Report

Comments and Suggestions for Authors

My first question is what kind of paper it is "Article" means a research article or a "Review"

The text appears to be well-written and professionally edited. However, there are a few minor stylistic inconsistencies, such as occasional variations in hyphenation (e.g., "plant-growth promoting" vs. "plant growth promoting"), which do not significantly impact the overall quality of the writing. 

Some of the minor editorials and critical questions:

Line 2-3: Please capitalize each word in the title, subtitle, and subheadings throughout the manuscript. 

Line 16 "Over" should be in regular font

Line 36 "Bacillus" should not be bold

Line 96: Limited database: The study relies solely on the Web of Science Core Collection database, which may not capture all relevant publications in the field.

The study focuses exclusively on Bacillus biofertilizers without comparing trends or effectiveness with other types of biofertilizers or traditional fertilizers.

As authors mentioned in section 4.4. Line 670: The study's reliance on keyword analysis may not fully capture the nuances of research topics, especially if authors use different terminologies for similar concepts.

Despite these limitations, the study provides a valuable contribution to understanding the development and current state of Bacillus biofertilizers research, offering useful insights for future research directions in this field.
